# Human Plasma Xanthine Oxidoreductase Activity in Cardiovascular Disease: Evidence from a Population-Based Study

**DOI:** 10.3390/biomedicines11030754

**Published:** 2023-03-01

**Authors:** Yuka Kotozaki, Mamoru Satoh, Takahito Nasu, Kozo Tanno, Fumitaka Tanaka, Makoto Sasaki

**Affiliations:** 1Iwate Tohoku Medical Megabank Organization, Iwate Medical University, 1-1-1, Idaidori, Yahaba 028-3694, Iwate, Japan; 2Department of Biomedical Information Analysis, Institute for Biomedical Sciences, Iwate Medical University, 1-1-1, Idaidori, Yahaba 028-3694, Iwate, Japan; 3Division of Cardiology, Department of Internal Medicine, Iwate Medical University, 2-1-1, Idaidori, Yahaba 028-3694, Iwate, Japan; 4Department of Hygiene and Preventive Medicine, Iwate Medical University, 1-1-1, Idaidori, Yahaba 028-3694, Iwate, Japan; 5Division of Nephrology and Hypertension, Department of Internal Medicine, Iwate Medical University, 2-1-1, Idaidori, Yahaba 028-3694, Iwate, Japan; 6Division of Ultrahigh field MRI, Institute for Biomedical Sciences, Iwate Medical University, 1-1-1, Idaidori, Yahaba 028-3694, Iwate, Japan

**Keywords:** plasma xanthine oxidoreductase, renal function, reactive oxygen species production, atherosclerosis, cardiovascular disease

## Abstract

Xanthine oxidoreductase (XOR) and its products contribute to the development of chronic inflammation and oxidative stress. Excessive XOR activity is believed to promote inflammatory responses and atherosclerotic plaque formation, which are major cardiovascular risk factors. The mechanisms of XOR activity in the development and progression of cardiovascular disease (CVD), coupled with the complexity of the relationship between XOR activity and the biological effects of uric acid; reactive oxygen species; and nitric oxide, which are the major products of XOR activity, have long been debated, but have not yet been clearly elucidated. Recently, a system for measuring highly sensitive XOR activity in human plasma was established, and there has been progress in the research on the mechanisms of XOR activity. In addition, there are accumulating findings about the relationship between XOR activity and CVD. In this narrative review, we summarize existing knowledge regarding plasma XOR activity and its relationship with CVD and discuss future perspectives.

## 1. Introduction

Xanthine oxidoreductase (XOR) is a member of the molybdenum enzyme family, which has gained increasing attention owing to its ability to generate reactive oxygen species (ROS), its suspected role in reperfusion injury, and more recently, its pathophysiological role in congestive heart failure [1]. XOR activity is recognized as a significant source of ROS [2], contributing to the development of oxidative stress-related tissue injury [2,3,4]. In addition, XOR activity is involved in vascular inflammation and the consequent development of atherosclerosis [5], thus demonstrating the close relationship between XOR activity and arteriosclerosis which is thought to involve various mechanisms.

The function of XOR has been examined primarily via studies of laboratory mice and rabbits. Historically, it has been difficult to accurately measure human plasma XOR activity because human XOR activity is lower than that of mice and other animals [6]. Another drawback of previous measurement methods of XOR activity (e.g., liquid chromatography/ultra-violet (LC/UV), fluorometry, and radiochemical assays) is that they are susceptible to endogenous substances such as hypoxanthine, xanthine, and UA [7,8,9,10]. In recent years, a sensitive method was developed to measure XOR activity in human plasma, unaffected by the original uric acid (UA) concentration in the sample, using stable isotope-labeled [^13^C_2_,^15^N_2_] xanthine and liquid chromatography/triple quadrupole mass spectrometry [10,11,12]. This method involved small molecules (e.g., hypoxanthine, xanthine, and UA) which were removed from plasma samples and mixed in Tris buffer with (^13^C_2_, ^15^N_2_)-xanthine and nicotinamide adenine dinucleotide (NAD^+^) substrates and (^13^C_3_, ^15^N_3_)-UA as an internal standard. The development of these measurement techniques has led to studies that elucidate the mechanism of plasma XOR activity in humans. 

This review focuses on the evidence linking human plasma XOR activity to cardiovascular disease (CVD). 

## 2. XOR and the Pathway by Which It Acts

XOR is an enzyme that catalyzes the final two steps of the purine degradation system, hydroxylating hypoxanthine to produce xanthine and hydroxylating xanthine to produce UA [13]. XOR has been shown to act only as a xanthine dehydrogenase (XDH) in non-mammalian organisms. Furthermore, XOR can be reversibly converted to XDH and xanthine oxidase (XO) in mammals [3]. 

The mammalian XOR protein has three domains: (1) molybdopterin cofactor containing a molybdenum atom (Moco), (2) flavin adenine dinucleotide (FAD), and (3) two non-identical iron–sulfur clusters (2Fe/S) [14]. The Moco site, the largest domain, contains the substrate pocket for XDH, XO, and nitrite reductase activity, respectively; the FAD site, the second largest domain, is responsible for nicotinamide adenine dinucleotide (NADH) oxidase activity. The electron (e−) flux moves from the Moco site, where oxidation occurs, to the FAD site, where two iron–sulfur redox products are reduced by electron acceptors. XOR has low substrate specificity and diverse activities, oxidizing and reducing many endogenous and exogenous products and functioning as a detoxification and drug metabolizing enzyme [15]. XOR is produced devoid of the molybdenum or sulfur, which can result in loss of enzymatic activity at the Moco site [3]. The mammalian XOR is constitutively an NAD^+^-dependent dehydrogenase and can be reversibly converted to an oxidase by oxidation of two cysteine residues or irreversibly by partial proteolysis of a fragment containing such a cysteine group [3]. To summarize, the four products of XOR activities are: (1) UA and reduced NADH from XDH, (2) UA and ROS from XO, (3) nitric oxide (NO) from nitrate and nitrite reductase, and (4) ROS from NADH oxidase [16]. The activities of XOR are schematized in Figure 1.

The mammalian XOR is constitutively an NAD^+^-dependent dehydrogenase, reversibly transformed into an oxidase by oxidation of two cysteine residues or irreversibly transformed by partial proteolysis of fragments containing such cysteine groups [17,18]. The transition from XDH to XO includes the intermediate XOR form, which engages in both dehydrogenase and oxidase activities, as only one of the two important sulfhydryl groups is oxidized [19]. The conversion from XDH to XO occurs by oxidation of the sulfhydryl groups when the enzyme is released again from the cell into the gastrointestinal tract lumen, urinary tract, or into biological fluids such as milk or serum [3]. The conversion from XDH to XO was observed to occur through oxidation (reversible) in a variety of hypoxic/ischemic and other pathological conditions. However, some prolonged ischemic conditions have also described a proteolytic conversion (irreversible) [20,21,22]. In addition, low oxygen tension induced by sickle cell disease (SCD) can result in significant release of XDH from the liver into circulation, thus significantly increasing the amount of serum XO [23].

The XO form catalyzes the transfer of monovalent and divalent electrons to O_2_, producing superoxide ions and hydrogen peroxide (H_2_O_2_), respectively. XDH can produce these ROS at the FAD site by acting as an NADH oxidase. This activity has been found to be retained even when the molybdenum or sulfur atoms of the molybdopterin cofactor are missing and when XOR is inhibited by competitive or non-competing drugs that block the function of the Moco site [3]. XOR, which has been demonstrated to function as a nitrate reductase that reduces nitrate to nitrite and as a nitrite reductase that reduces nitrite to NO, is known to reduce nitrate to nitrite at the Moco site [24].

XOR catalyzes the last two steps of purine catabolism, because of the lack of uricase that has been lost during evolution, in humans and other higher primates. Therefore, resulting in hyperuricemia in these primates more than in uremic mammals [25]. XOR has the rate-limiting function of generating irreversible products, xanthine and UA, precluding the purine nucleotide salvage pathway [26]. Serum XOR is mainly derived from physiological hepatic cell turnover, which induces the release of liver enzymes from dead cells into circulation. The serum XOR level strongly increases due to several liver pathologies that cause tissue damage [15].

The production of human XOR protein is tightly regulated at both the transcriptional and post-translational levels. The basic activity of the human XOR promoter is low compared to other mammals. The reason may be due to repressor elements identified in non-coding regions of the XOR [27]. The human XOR gene (hXOR) expression is usually downregulated in almost all tissues except breast epithelial cells during lactation, the gastrointestinal tract, the kidney, and the liver [28]. The breast’s secretion of lactating milk fat globules is carried out by clustering the transmembrane protein butyrophilin1A1 (Btn). XOR induces apical membrane reorganization, which allows Btn clustering and membrane docking of milk fat droplets, contributing to the secretion of apocrine lipids [29]. In the gut, XOR is abundant in the intestinal lumen because of the frequent replacement of intestinal cells. As a result, XOR-derived ROS and reactive nitrogen species (RNS) protect against opportunistic infections, and indigenous bacteria are protected [30]. Catabolism of most purines occurs in the kidney. Therefore, in this region, XOR is responsible for uricosuria and, consequently, for uricemia levels, which contribute to supporting blood pressure by upregulating cyclooxygenase-2 (COX-2) expression and, consequently, the renin/angiotensin pathway [31]. In the liver, XOR performs all its activities and metabolizes many endogenous and exogenous substrates, including drugs and purines. UA produced by XOR affects hepatic glucose and lipid metabolism, increasing gluconeogenesis and fat accumulation [32]. Serum XOR is mainly derived from physiological hepatocyte turnover, which induces the release of liver enzymes from dead cells. Moreover, circulating XOR can bind to endothelial cells, promote endothelial cell activation during inflammation, regulate vascular tone, and consequently contribute to blood pressure regulation [33].

Several factors, such as hypoxia, cytokines, growth factors, and various hormones, can elevate hXOR expression and enzymatic activity through both transcriptional upregulation and post-translational activation of XOR [3,34,35]. This explains why XOR activity is elevated during hypoxic/ischemic conditions and inflammation.

## 3. XOR Activity and Vascular Injury

XDH reduces NAD^+^ to NADH, and XO uses oxygen to produce H_2_O_2_ and superoxide [36,37]. Insufficient activation of XOR and XO results in increased ROS and increased oxidative stress [37]. ROS is known for its role in lipid, protein, and DNA damage [38]. Oxidative stress may contribute significantly to endothelial dysfunction in CVD because superoxide radicals readily inactivate endothelial NO [39].

XOR is released from hepatocytes into the circulatory system, where it is converted from dehydrogenase to oxidase, and binds with high affinity to the glycosaminoglycans on the endothelium surface [4,31]. The XOR bound to endothelial cells acts as a regulator of systemic redox balance and sets up several important endothelial functions [31,40]. In addition, endothelium-linked XOR generates ROS and RNS, which activate endothelial cells and contribute to their permeability and exudate formation [37,41]. 

XOR-derived NO, H_2_O_2_, and superoxide ions have pro-inflammatory activity because they increase vascular endothelial permeability and regulate local vasodilation [36]. XOR upregulates signaling molecules that promote cell proliferation, migration, and tissue repair [28]. XOR is also an endothelial enzyme that regulates NO concentration, which is important for the regulation of arteriolar vascular tone and thus blood pressure [21,31].

Endothelium-bound XOR inhibits NO-dependent cyclic guanosine monophosphate (cGMP) production in smooth muscle cells, contributing to impaired vasorelaxation [42].

Oxidative stress has pro-inflammatory effects and contributes to the oxidation of low-density lipoprotein (LDL) particles and the formation of advanced glycation end products (AGEs), which are associated with the atherosclerotic process [37]. Atherosclerosis is promoted by oxidative stress that induces oxidative modification of circulating molecules, as well as endothelial cell and monocyte/macrophage activation and smooth muscle cell proliferation [43]. Oscillatory shear stress increases endothelial XOR expression and activity, activates nicotinamide adenine dinucleotide phosphate (NADPH) oxidase, and significantly increases ROS production by promoting the conversion of XDH to XO, thereby promoting atherosclerosis [44]. ROS, generated by NADPH oxidase and XO, oxidize and inactivate tetrahydrobiopterin, causing the uncoupling of endothelial nitric oxide synthase (eNOS), owing to its cofactor deficiency that increases oxidative stress [45]. Increasing evidence supports the hypothesis that elevated plasma XOR activity contributes to the development of atherosclerosis and plays an important role in the pathogenesis of CVD [46]. 

A pathophysiological model of heart failure (HF) with preserved ejection fraction (HFpEF) was developed, proposing advanced comorbidities of HF such as aging, diabetes, metabolic syndrome, salt-sensitive hypertension, and atrial fibrillation. In addition, the model hypothesizes that these diseases have detrimental effects on the heart via the endothelium in the coronary microcirculation, which acts as a sort of central processor, transmitting damage to the heart [47]. An upregulation of XOR activity is present in many of these comorbidities, which act via the endothelium in the coronary microcirculation and induce inflammatory conditions, thereby stimulating endothelial cells to produce ROS [48].

Atherosclerotic plaques after carotid endarterectomy, cholesterol, and XOR have been found to co-localize. An increase in the local UA concentration in such plaques has also been found, suggesting an upregulation of XOR activity [49]. Patients with ischemic encephalopathy have significantly higher expression of XOR in the macrophages of carotid atherosclerotic plaques; in addition, the overexpression of XOR activity in macrophages is associated with low levels of high-density lipoprotein (HDL) and co-occurs with the formation of cholesterol crystals in atheroma and the inflammatory response of macrophages [50].

## 4. Plasma XOR Activity and Cardiovascular Events in Patients with CVD

XOR-derived ROS and oxidative stress may cause endothelial dysfunction that promotes the onset and progression of atherosclerosis and leads to CVD development [51]. A cross-sectional study of 42 children and adolescents showed that serum XOR activity was significantly higher in obese children than in healthy-weight children; in addition, it was significantly positively correlated with body mass index (BMI) z-score, waist circumference, and serum levels of oxidized LDL, and negatively correlated with monocyte chemoattractant protein-1, adiponectin, and HDL, suggesting that XOR activity may play a role in increased cardiovascular risk [52]. A follow-up study of 12 morbidly obese patients undergoing bariatric surgery found that changes in aspartate aminotransferase (AST) and alanine aminotransferase (ALT), but not BMI, were significantly positively associated with changes in plasma XOR activity both one week and one year after surgery. There was also no significant correlation between changes in plasma XOR activity and changes in plasma uric acid levels. These results suggest that increased plasma XOR activity was primarily associated with liver dysfunction, such as nonalcoholic fatty liver disease/steatohepatitis, rather than obesity per se [53].

Patient studies are underway to investigate the mechanisms of plasma XOR activity. In a prospective study of 301 outpatients with CVD, obese cardiovascular patients with diabetes were independently associated with higher plasma XOR activity, which was also associated with higher plasma hydrogen peroxide (H_2_O_2_) levels [54]. 

Patients with HF have been found to have markedly reduced endothelium-associated superoxide dismutase (SOD) activity and over 200% greater endothelium-bound XOR activity compared to controls [55]. A prospective study of patients with acute HF showed that patients in the low XOR group had significantly worse prognoses than those in the high XOR group within 365 days. In addition, a decrease in the XOR activity level within 14 days after admission due to HF treatment has a positive prognostic significance, whereas an insufficient reduction in XOR activity is associated with an increase in further HF events [56]. A 3-year follow-up study of 484 patients showed a U-shaped association between abnormally high or low plasma XOR activity and the severity of chronic HF and clinical outcome after adjusting for confounding risk factors [57]. A prospective study of 257 patients with HFpEF with a median follow-up of 809 days showed that regardless of hyperuricemia status, high XOR activity was significantly associated with and an independent risk factor for major adverse cardiovascular events after adjusting for confounding factors [58]. 

A population-based study of patients with CVD showed that the plasma XOR activity level was significantly positively correlated with BMI, serum liver enzyme level, and glycated hemoglobin (HbA1c). The plasma XOR activity level also showed a significant negative correlation with renal function and left ventricular hypertrophy and a U-shaped correlation with ventricular ejection fraction and elevated serum level of type B natriuretic peptide (BNP) [59]. 

A study from Japan found that plasma XOR activity was significantly higher in hemodialysis patients with diabetes than in those without diabetes; in addition, plasma glucose and serum HbA1c were significantly and independently associated with plasma XOR activity in patients with diabetes, and serum UA was significantly and independently associated with plasma XOR activity in those without diabetes. This means that glycemic control in hemodialysis patients may be important for decreasing ROS induced by XOR activity [60]. 

Moreover, UA, a product of XOR, has various physiological roles, including acting on the kidneys to support blood pressure and being a pro-inflammatory substance; in addition, it affects hepatic metabolism by promoting gluconeogenesis and fat accumulation [61] and has been implicated in CVD development [46]. Hyperuricemia accompanies metabolic syndrome, hypertension, diabetes, dyslipidemia, chronic renal disease, and obesity [62]. Hyperuricemia significantly increases the rate of CVD [63]. In vitro experiments suggest a possible mechanism of endothelial dysfunction and consequent kidney disease owing to the phenotypic transition of vascular endothelium induced by UA through oxidative stress and glycocalyx shedding [64]. Furthermore, high XOR activity and hyperuricemia are risk factors for chronic kidney disease (CKD) and CVD development [37].

A study of 118 patients with CKD showed that estimated glomerular filtration (eGFR) and levels of HbA1c or plasma glucose were significantly, independently, and positively associated with plasma XOR activity after adjusting for several confounders; in addition, plasma XOR activity was significantly higher in diabetic CKD patients than in those without diabetes, and the relationship between glycemic control and plasma XOR activity was significant even in CKD patients without uric acid-lowering drug treatment. The findings of this study suggest that glycemic control in CKD patients in regard to decreased XOR is important and could possibly lead to a decrease in CVD events [65]. A prospective study of 101 patients with CKD or on chronic hemodialysis who were followed for up to three years showed that the serum level of XOR activity, but not UA, was an independent predictor of cardiovascular events. This result suggests that XOR activity plays a role in CVD related to CKD by inducing oxidative stress [66]. Table 1 describes the studies presented in this section in more detail.

## 5. Plasma XOR Activity and Cardiovascular Events in the General Population

There is a growing body of knowledge, including large cohort studies, on the role of plasma XOR activity in the general population. A cross-sectional study of 156 participants registered in the health examination registry showed that plasma XOR activity may contribute to the pathophysiology of high blood pressure through ROS but not UA production, especially in patients with lower oxidative stress [67].

A population-based cohort study of 627 Japanese participants confirmed that plasma XOR activity was significantly higher in males than in females and that habitual smoking was associated with elevation of plasma XOR activity. In addition, BMI, smoking, and levels of alanine transaminase, UA, triglycerides, and homeostasis model assessment of insulin resistance (HOMA-R) were independent predictors of plasma XOR activity after adjusting for age and gender. The results of this cross-sectional study suggest that plasma XOR activity is a novel biomarker of metabolic disorders in the general population [68].

Our community-based cohort study of 1631 Japanese participants confirmed that plasma XOR activity levels were independently associated with BMI, diabetes mellitus, dyslipidemia, and UA levels. In addition, the top quartile of XOR activity was associated with a higher risk of CVD, calculated using the Framingham risk score, after adjusting for baseline characteristics. The results of our cross-sectional study suggest that high XOR activity is a marker of cardiovascular risk, at least in the general Japanese population [69]. 

Furthermore, our previous community-based cohort study of 4248 Japanese participants confirmed that plasma XOR activity levels were significantly higher in the group with CKD stage G3 and G4. In addition, increased plasma XOR activity was associated with a higher CKD stage after adjusting for baseline characteristics. This result suggests that high XOR activity is associated with the severity of CKD, at least in the general Japanese population, suggesting that upregulated XOR activity may be involved in advanced renal dysfunction [70]. The studies presented in this section are summarized in Table 2.

Figure 2 shows the relationship between plasma XOR activity and CVD, which is becoming clearer through clinical studies. These results indicate that XOR activity may be a more reliable biomarker of metabolism, CVD, and renal function [67,68,69]. Studies on the role of XOR activity in the development and progression of CVD have only recently begun, but the mechanisms are slowly becoming clearer. Future studies on how plasma XOR activity contributes to the onset and progression of CVD are warranted.

Elevated plasma XOR activity increases ROS. Oxidative stress is increased, inflammation is induced, and the vascular endothelium is easily damaged. In addition, arteriosclerosis progresses by promoting plaque formation. Therefore, CVD may occur more easily. In addition, there is growing evidence from studies in Japanese populations that plasma XOR activity may be a more reliable biomarker of CVD.

## 6. Conclusions

In this review, we summarized recent progress in the understanding of the evidence linking human plasma XOR activity to CVD. Although XOR activity in mammals has been measured in the past, the problem is that human plasma XOR activity is lower than in other animals, making it difficult to measure accurately. However, improved methods for measuring human plasma XOR activity have quickly accelerated the study of XOR kinetics in patients and the healthy general population over the past few years. The relationship between XOR activity and CVD has been studied for several years, and knowledge has been accumulated, but it is still a topic of ongoing debate. One reason for this is the complex relationship between XOR and its products. To improve our understanding of the dynamics and role of plasma XOR activity in the progression of CVD, it is important to consider these processes more deeply, including the impact the activity of XOR and its products has on them.

Elevated XOR activity is responsible for the formation of UA from hypoxanthine and xanthine, leading to superoxide and ROS production [71,72]. ROS production in the vessel wall is involved in the progression of arteriosclerosis [73]. XOR activity is thought to reflect the degree of progression of atherosclerosis because it generates ROS and causes endothelial dysfunction [3,36,46]. In addition, elevated XOR activity and relative production of UA and ROS have been suggested to promote the early stages of CKD via their effects on microcirculation, tissue damage, and the resulting effects on the microarteries and the promotion of hypertension [74,75].

XOR activity has long been studied based on results obtained from animal studies. The role and dynamics of plasma XOR activity have been examined in human studies only in the last 5–6 years. As a result, we and another research group suggested that in Japanese populations, plasma XOR activity may be used as a biomarker to assess metabolism, renal function, and CVD progression [68,69,70]. In addition, plasma XOR activity was recently suggested as a possible risk factor for CVD [76]. It is necessary to confirm if this is true in other populations as soon as possible.

To better understand the dynamics of plasma XOR activity, it is necessary to examine longitudinal changes in plasma XOR activity and the relationship between plasma XOR activity and other physiological measurements such as flow-mediated dilation and brachial-ankle pulse wave velocity. Furthermore, it is important to examine the association of plasma XOR activity with markers of CVD development, such as myocardial infarction and stroke, using disease registry data. The investigation of these associations will provide new perspectives into the mechanisms of plasma XOR activity and CVD development. 

## Figures and Tables

**Figure 1 biomedicines-11-00754-f001:**
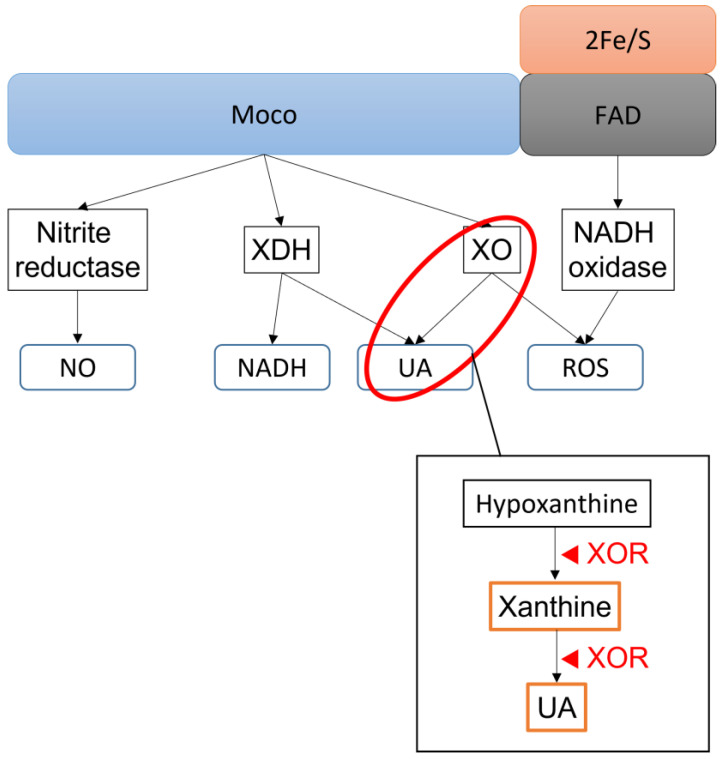
XOR activities and products.

**Figure 2 biomedicines-11-00754-f002:**
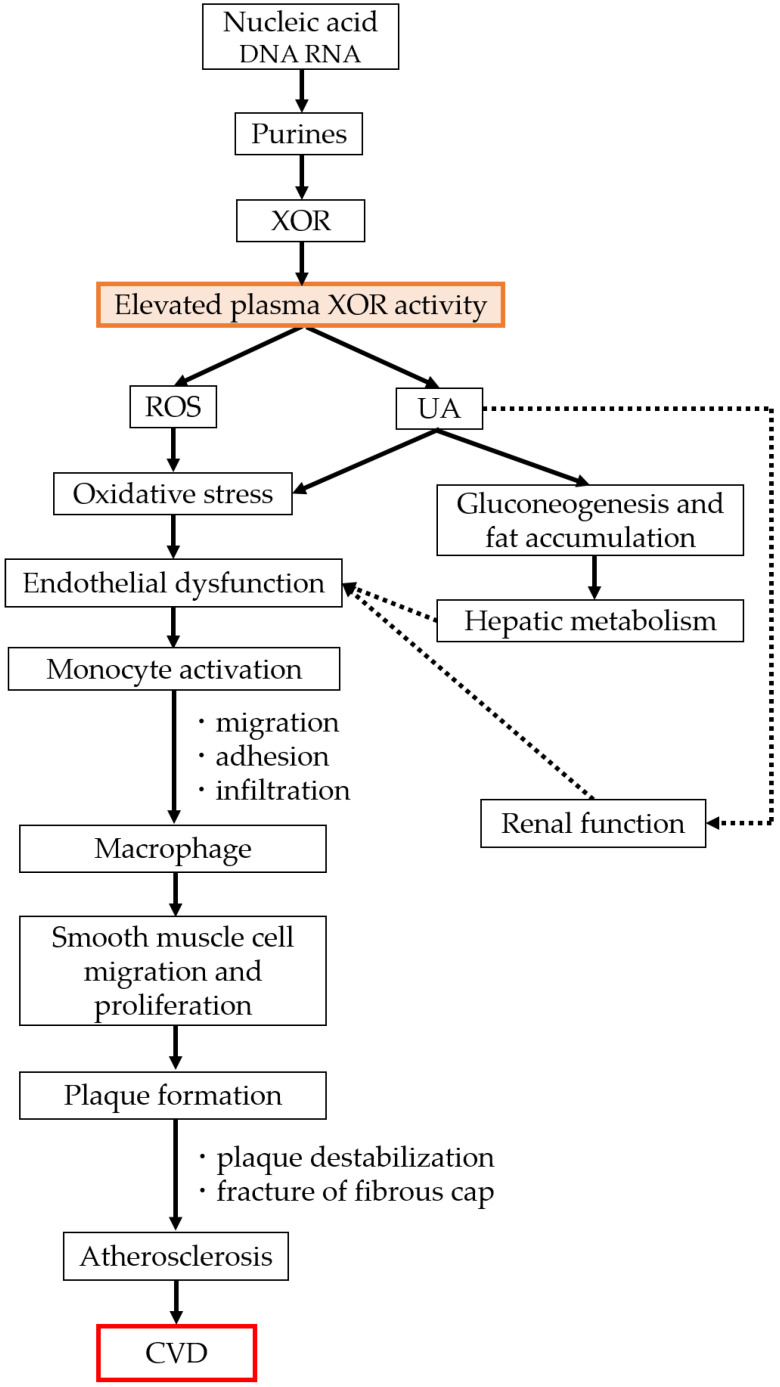
The relationship between plasma XOR activity and CVD.

**Table 1 biomedicines-11-00754-t001:** Summary of the studies on patients with CVD.

Authors	Year	Study Group	Results
Tam H.K. et al. [52]	2014	42 children and adolescents	A 3.8-fold increase in plasma XOR activity in obese children compared to healthy-weight children (*p* < 0.001). Plasma XOR activity was correlated with BMI z-score (r = 0.41), waist circumference (r = 0.41), HDL cholesterol (r = −0.32), oxidized LDL (r = 0.57), adiponectin (r = −0.53), and monocyte chemotactic protein-1 (r = −0.59).
Kawachi Y. et al. [53]	2021	12 morbidly obese patients who underwent bariatric surgery	One year after bariatric surgery, plasma XOR activity decreased significantly (*p* = 0.0001). Changes in AST and ALT, but not in BMI, had significant positive associations with changes in plasma XOR activity both 1 week (AST: r = 0.94; *p* < 0.0001, ALT: r = 0.91; *p* < 0.0001) and 1 year (AST: r = 0.76; *p* = 0.004, ALT: r = 0.80; *p* = 0.002) after the surgery.
Matsushita M. et al. [54]	2020	301 outpatients with CVD	Obese cardiovascular patients with diabetes were independently associated with higher plasma XOR activity (OR, 2.683; 95% CI: 1.441–4.996). BMI was independently associated with high plasma XOR activity (OR, 1.340; 95% CI: 1.149–1.540). Plasma hydrogen peroxide was significantly higher in diabetes patients with high plasma XOR activity and obesity (>22 kg/m^2^) than in other patients (median, 2967 vs. 2725 relative fluorescence unit).
Landmesser U. et al. [55]	2002	14 patients with CHF and 10 control participants	Patients with HF were found to have markedly reduced endothelium-associated SOD activity (5.0 ± 0.7 vs. 14.4 ± 2.6 U mL^−1^ min^−1^; *p* < 0.01). Patients with HF had over 200% more endothelium-bound XOR activity compared to controls (38 ± 10 vs. 12 ± 4 nmol O_2_^·−^ μL^−1^; *p* < 0.05).
Okazaki H. et al. [56]	2019	118 AHF patients and 231 controls who attended a cardiovascular outpatient clinic	Plasma XOR activity in the AHF group was significantly higher than that in the control group (104.0 vs. 45.2 pmol/h/mL; *p* < 0.001). Serum UA (per 1.0 mg/dL increase, OR: 1.28; 95% CI: 1.07–1.54; *p* = 0.008) and lactate levels (per 1.0 mmol/L increase, OR: 1.24; 95% CI: 1.04–1.48; *p* = 0.016) were independently associated with high plasma XOR activity during the acute phase of AHF.
Otaki Y. et al. [57]	2017	440 patients with CHF and 44 control participants	Both high and low levels of XOR activity were significantly associated with cardiac events in patients with CHF after adjusting for confounding risk factors, including serum UA and loop diuretic use (both *p* < 0.0001). The cardiac event rate was significantly higher in patients with either high or low XOR activity (high XOR activity: *p* = 0.0007; low XOR activity: *p* = 0.0729). The net reclassification index was significantly improved by adding XOR activity to the basic risk factors (C-index = 0.807; *p* = 0.0006).
Watanabe K. et al. [58]	2020	257 patients with HFpEF	High XOR activity was significantly associated with major adverse cardiovascular events (MACEs) after adjusting for confounding factors (HR: 3.6; 95% CI: 1.68–8.12; *p* < 0.001). High XOR activity was associated with MACEs, regardless of the hyperuricemia status (HR: 4.0 to 4.4; *p* < 0.001).
Fujimura Y. et al. [59]	2017	270 patients with CVD without UA-lowering drug treatment	XOR activity was associated with BMI (*p* = 0.004), ALT (*p* < 0.001), HbA1c (*p* < 0.001) and renal function (*p* < 0.001). Compared with patients with the lowest XOR activity quartile, those with the higher three XOR activity quartiles more frequently had left ventricular hypertrophy (*p* = 0.031, by χ^2^ test). Plasma XOR activity showed a U-shaped association with low left ventricular ejection fraction (*p* = 0.071) and increased plasma B-type natriuretic peptide levels (*p* < 0.001), and these associations were independent of age, gender, BMI, ALT, HbA1C, serum UA, and CKD stage.
Nakatani A. et al. [60]	2017	163 hemodialysis patients	Plasma glucose (r = 0.23, *p* = 0.003) and serum UA levels (r = 0.23, *p* = 0.003) correlated significantly and positively with plasma XOR activity. Diabetes (β = 0.16, *p* = 0.028) and plasma glucose (β = 0.30, *p* < 0.001) were significantly, independently, and positively associated with plasma XOR activity. Serum UA (r = 0.29, *p* = 0.007) significantly and positively correlated with plasma XOR activity in hemodialysis patients without diabetes; plasma glucose (r = 0.34, *p* = 0.003) and serum glycated albumin (r = 0.29, *p* = 0.015) significantly and positively correlated with plasma XOR activity in those with diabetes. Serum UA was significantly and independently associated with plasma XOR activity in those without diabetes (β = 0.20, *p* = 0.042).
Nakatani S. et al. [65]	2021	118 pre-dialysis CKD patients	eGFR (β = 0.22, *p* = 0.028) and HbA1c (β = 0.33, *p* < 0.001) were significantly, independently, and positively associated with plasma XOR activity after adjusting for several confounders (R^2^ = 0.26, *p* < 0.001), as were eGFR (β = 0.28, *p* = 0.007) and plasma glucose (β = 0.25, *p* = 0.007) (R^2^ = 0.22, *p* < 0.001) Plasma XOR activity was significantly higher in CKD patients with diabetes than in those without diabetes (62.7 vs. 25.7 pmol/h/mL, *p* < 0.001). The association between glycemic control and plasma XOR activity was significant even in CKD patients without uric acid-lowering drug treatment (R^2^ = 0.44, *p* < 0.001).
Gondouin B. et al. [66]	2015	51 CKD and 50 hemodialysis patients were compared to 38 matched healthy controls	XOR activity was negatively correlated with SOD (r = −0.2, *p* = 0.04) and positively correlated with malondialdehyde (r = 0.3, *p* = 0.004). XOR activity was an independent predictor of cardiovascular events in CKD and hemodialysis patients, regardless of UA levels (HR: 1.55; 95% CI: 1.09–2.19; *p* < 0.05).

**Table 2 biomedicines-11-00754-t002:** Summary of the studies discussed in the general population.

Authors	Year	Study Group	Results
Yoshida S. et al. [67]	2020	156 participants registered in the health examination registry	Plasma XOR activity, but not serum UA, was significantly associated with mean arterial pressure (MAP) (β = 0.21, *p* = 0.019). Plasma XOR activity was shown to be significantly and positively associated with MAP in patients with a lower derivative of reactive oxygen metabolite levels (β = 0.43, *p* < 0.001).
Furuhashi M. et al. [68]	2018	627 participants registered in a population-based cohort	Plasma XOR activity was significantly higher in males than in females (43 vs. 32 pmol/h/mL plasma, *p* = 0.002), and habitual smoking was associated with elevation of activity (*p* < 0.05). BMI (r = 0.32, *p* < 0.001), waist circumference (r = 0.29, *p* < 0.001), levels of liver enzymes, including alanine transaminase (r = 0.69, *p* < 0.001), UA (r = 0.25, *p* < 0.001), triglycerides (r = 0.31, *p* < 0.001), and HOMA-IR (r = 0.24, *p* < 0.001), were independent predictors of plasma XOR activity after adjusting for age and gender.
Kotozaki Y. et al. [69]	2021	1631 participants registered in a community-based cohort	Plasma XOR activity was significantly higher in males than in females (43.7 vs. 31.6 pmol/h/mL plasma, *p* < 0.001). Plasma XOR activity was independently associated with BMI (β = 0.26, *p* < 0.001), diabetes (β = 0.09, *p* < 0.001), dyslipidemia (β = 0.08, *p* = 0.001), and UA (β = 0.13, *p* < 0.001). The highest quartile of plasma XOR activity was associated with a high risk for CVD (Framingham risk score ≥15) after adjusting for baseline characteristics (OR, 2.93; 95% CI: 1.16–7.40). The area under the receiver operating characteristic curve of the Framingham risk score with XOR activity was 0.81 (*p* = 0.008).
Taguchi S. et al. [70]	2022	4248 participants registered in a community-based cohort	Blood pressure, BMI, UA, LDL cholesterol, and HbA1c were highest in the highest plasma XOR quartile (all *p* < 0.001). Plasma XOR activity was significantly higher in the subgroup with CKD stage G3 and G4 (G1 vs. G2 vs. G3–G4: 44.8 ± 40.5 vs. 52.0 ± 42.9 vs. 54.1 ± 43.9 pmol/h/mL, *p* = 0.02). The odds ratios (95% CIs) per 1 pmol/h/mL increase in XOR activity with CKD stage G1 as a reference were 1.37 (1.13–1.73) in G2 and 1.51 (1.30–1.84) in G3–G4.

## Data Availability

Not applicable.

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
