# Peer review of "Human Plasma Xanthine Oxidoreductase Activity in Cardiovascular Disease: Evidence from a Population-Based Study"

_biomedicines, 2023, doi:10.3390/biomedicines11030754_

Round 1
Reviewer 1 Report
Abstract
Please indicate if this is a systematic or narrative review
1-Introduction
The introduction is a bit too short. Specify other techniques for measuring XOR with the problems encountered and indicate why the one you indicate seems to be the most appropriate
2-XOR activity and vascular injury
This section should be refocused on the pathophysiology of vascular injury induced by increased XOR activity. This section contains information that should be included in the section below.
3-Plasma XOR activity and cardiovascular events in patients with CVD
This part should be synthetic in order to avoid the same turns of phrase "A cross-sectional study".
4-Plasma XOR activity and cardiovascular events in the general population
What do you mean by general population?
The data presented in this section concern patients with cardiometabolic disorders
Figure 1 is very interesting and could represent a framework for redesigning your manuscript, but is an alteration of XOR activity a cause or consequence of CVD?
Conclusion
You mentioned that “The relationship between XOR activity and 204 CVD has been studied for several years, however, the results have not been satisfactory to many researchers”, however, the majority of the studies presented show a relationship between increased XOR activity and CVD
Author Response
Thank you very much for providing important insights. We are grateful for the time and energy you expended on our behalf.
Abstract: Please indicate if this is a systematic or narrative review.
Response: Thank you for your comment. We have added “narrative review as a type of review (see page 1, line 27).
- Introduction: The introduction is a bit too short. Specify other techniques for measuring XOR with the problems encountered and indicate why the one you indicate seems to be the most appropriate.
Response: Thank you for your suggestion. We have incorporated some comments and added some citations concerning other techniques for measuring XOR with problems into the introduction section of a revised manuscript (see page 1, lines 42-46; and page 2, lines 50-53).
- XOR activity and vascular injury: This section should be refocused on the pathophysiology of vascular injury induced by increased XOR activity. This section contains information that should be included in the section below.
Response: As a reviewer’s comment, we have added some comments that focused on the pathophysiology of vascular injury induced by increased XOR activity into “XOR activity and vascular injury” section (see page 4, lines 139-189).
- Plasma XOR activity and cardiovascular events in patients with CVD: This part should be synthetic in order to avoid the same turns of phrase "A cross-sectional study".
Response: As a reviewer’s comment, we have modified from “A cross-sectional study” to “A study” (see page 6, lines 230-236, and lines 248-259).
- Plasma XOR activity and cardiovascular events in the general population: What do you mean by general population? The data presented in this section concern patients with cardiometabolic disorders.
Response: Thank you for the valuable suggestions. We summarize the relationship between XOR and CVD examined using patient data in Section 3 and using cohort data from local residents in Section 4. We thought your question was based on the results of the study by Tam et al., and checked the study again and it was on patients. Therefore, we have moved the description of the study by Tam et al. to a third section (see page 5, lines 193-198, Table 1 and citation 52).
Also, during the English proofreading, a participant had converted a patient word. So, we rewrote the description to the correct word (see page 5, lines 264 and 281, Table 2).
- Figure 1 is very interesting and could represent a framework for redesigning your manuscript, but is an alteration of XOR activity a cause or consequence of CVD?
Response: Thank you for your question. We believe that increased XOR activity is a possible cause of CVD. We suspect that increased oxidative stress and inflammatory responses triggered by elevated plasma XOR activity may lead to increased atherosclerosis and consequently to increased susceptibility to CVD. However, we thought you were wondering about the pathway where XOR in the figure we created. Therefore, we have made modifications to the figure 2.
- Conclusion: You mentioned that “The relationship between XOR activity and 204 CVD has been studied for several years, however, the results have not been satisfactory to many researchers”, however, the majority of the studies presented show a relationship between increased XOR activity and CVD.
Response: Thank you for pointing this out. We wanted to show that the relationship between XOR and CVD is a topic that has been reported but is still being debated. So, we rewrote the description to the correct word (see page 12, lines 312-313).

Reviewer 2 Report
Kotozaki and collagues have summarized the correlation between XOR activity and CVD diseases in human studies. Overall this is a well-designed and interesting review. The factors at play are sufficiently presented and are adequately explained for a general audience of non-XOR experts. I have only few, but nevertheless important, suggestions to further improve the paper. 1) to add recent pubblished paper of Kawachi et al., JCI 2021.
2) to add in the first part of the review, after the introduction a paragraph concerning the enzyme XOR and also a figure of the pathway where XOR acts
Author Response
Response to Reviewer #2:
We thank the reviewer for their thoughtful assessment of our work and have revised the manuscript to address their comments.
- Kotozaki and collagues have summarized the correlation between XOR activity and CVD diseases in human studies. Overall this is a well-designed and interesting review. The factors at play are sufficiently presented and are adequately explained for a general audience of non-XOR experts. I have only few, but nevertheless important, suggestions to further improve the paper. 1) to add recent published paper of Kawachi et al., JCI 2021. 2) to add in the first part of the review, after the introduction a paragraph concerning the enzyme XOR and also a figure of the pathway where XOR acts.
Response: Thank you for the valuable suggestions. We have added the study of Kawachi et al. (see page 5, lines 198-205, Table 1 and citation 53).
- to add in the first part of the review, after the introduction a paragraph concerning the enzyme XOR and also a figure of the pathway where XOR acts
Response: we add after the introduction a paragraph concerning the enzyme XOR and also a figure of the pathway where XOR acts (see page 2, lines 57-138, and Figure 1).

Round 2
Reviewer 1 Report
The manuscript has been greatly improved and the answers provided by the authors are satisfactory